

# Unimodal relationship between small-scale barnacle recruitment and the density of pre-existing barnacle adults

Ricardo A. Scrosati and Julius A. Ellrich

Department of Biology, St. Francis Xavier University, Antigonish, Nova Scotia, Canada

## ABSTRACT

Recruitment is a key demographic process for population persistence. This paper focuses on barnacle (*Semibalanus balanoides*) recruitment. In rocky intertidal habitats from the Gulf of St. Lawrence coast of Nova Scotia (Canada), ice scour is common during the winter. At the onset of intertidal barnacle recruitment in early May (after sea ice has fully melted), mostly only adult barnacles and bare substrate are visible at high elevations in wave-exposed habitats. We conducted a multiannual study to investigate if small-scale barnacle recruitment could be predicted from the density of pre-existing adult barnacles. In a year that exhibited a wide adult density range (ca. 0–130 individuals $dm^{-2}$), the relationship between adult density and recruit density (scaled to the available area for recruitment, which excluded adult barnacles) was unimodal. In years that exhibited a lower adult density range (ca. 0–40/50 individuals $dm^{-2}$), the relationship between adult and recruit density was positive and resembled the lower half of the unimodal relationship. Overall, adult barnacle density was able to explain 26–40% of the observed variation in recruit density. The unimodal adult–recruit relationship is consistent with previously documented intraspecific interactions. Between low and intermediate adult densities, the positive nature of the relationship relates to the previously documented fact that settlement-seeking larvae are chemically and visually attracted to adults, which might be important for local population persistence. Between intermediate and high adult densities, where population persistence may be less compromised and the abundant adults may limit recruit growth and survival, the negative nature of the relationship suggests that adult barnacles at increasingly high densities stimulate larvae to settle elsewhere. The unimodal pattern may be especially common on shores with moderate rates of larval supply to the shore, because high rates of larval supply may swamp the coast with settlers, decoupling recruit density from local adult abundance.

# INTRODUCTION

Recruitment is a key demographic process that affects the persistence of populations. Thus, ecological research has often aimed at understanding its drivers (*Caley et al., 1996*; *Beck et al., 2001*; *Palumbi & Pinsky, 2014*). As barnacles are abundant organisms on seashores worldwide, they have often been used as model systems to study recruitment (*Jenkins et al., 2000*; *Navarrete, Broitman & Menge, 2008*; *Lathlean, Ayre & Minchinton, 2013*;

Corresponding author
Ricardo A. Scrosati, rscrosat@stfx.ca

*Menge et al., 2015*; *Barbosa et al., 2016*). For barnacles, settlement refers to the permanent contact with the substrate established by pelagic larvae (*Jenkins et al., 2000*), while recruitment is the appearance of new organisms in a benthic habitat that have resulted from the metamorphosis of settled larvae (*Ellrich et al., 2016a*). At regional spatial scales, common drivers of barnacle recruitment are seawater temperature and pelagic food (phytoplankton) supply, as both factors influence the growth and survival of the pelagic larvae (*Menge & Menge, 2013*; *Rognstad, Wethey & Hilbish, 2014*; *Scrosati & Ellrich, 2016*). At small spatial scales, factors such as substrate rugosity (*Coombes et al., 2015*), water motion (*Bertness et al., 1992*; *Ellrich & Scrosati, 2016*), macroalgal canopies (*Beermann et al., 2013*), and predator chemical cues (*Johnson & Strathmann, 1989*; *Ellrich, Scrosati & Molis, 2015*) influence barnacle recruitment in a variety of ways.

The presence of benthic conspecifics also influences barnacle recruitment at small scales. For instance, experiments conducted under laboratory (*Crisp & Meadows, 1962*; *Matsumura et al., 2000*; *Matsumura & Qian, 2014*) and field (*Chabot & Bourget, 1988*; *Raimondi, 1988*; *Jarrett, 1997*) conditions have shown that chemical and visual cues from adult barnacles attract pelagic conspecific larvae that are seeking settlement. Such an attraction is believed to aid larvae to locate adequate areas for growth and reproduction (*Clare, 2011*). In agreement with those findings, field experiments found that the presence of adult barnacles in moderate densities enhance barnacle recruitment at small spatial scales (*Bertness et al., 1992*; *Kent, Hawkins & Doncaster, 2003*; *Beermann et al., 2013*; *Ellrich et al., 2016a*).

The above studies suggest that small-scale recruitment in barnacle populations could be predicted from the density of pre-existing adult barnacles. Between low and intermediate adult densities, larval attraction by conspecific adults should result in a positive relationship between adult and recruit density. However, between intermediate and high adult densities, the abundant conspecific cues might indicate to pelagic larvae the potential for detrimental intraspecific interactions, as the resulting recruits might be crushed by growing adults or experience a reduced food supply outcompeted by the abundant adults (*Bertness, Gaines & Yeh, 1998*; *Hooper & Eichhorn, 2016*). In such conditions, recruit density (scaled to the substrate area available for settlement) could decrease with adult density because of larval repulsion. Therefore, for a wide range of adult density spanning low to high values, the adult–recruit relationship might be unimodal. This paper tests this hypothesis using field data from rocky intertidal habitats from Atlantic Canada.

## MATERIALS AND METHODS

We measured barnacle adult and recruit density at Sea Spray (45°46.4′N, 62°8.7′W), on the southern coast of the Gulf of St. Lawrence, Nova Scotia. This is a long-term reference location where we have monitored barnacles for several years (*Scrosati & Ellrich, 2016*). We surveyed habitats that face open waters, which makes these habitats wave-exposed, with daily values of maximum water velocity ranging between 4–8 m s$^{-1}$ (*Scrosati & Heaven, 2007*). The substrate is volcanic bedrock with a homogeneous rugosity and slope. On this coast, *Semibalanus balanoides* is the only species of intertidal barnacle

(*Scrosati & Heaven, 2007*). In Atlantic Canada, this species mates in autumn, broods in winter, and releases larvae to the water column in spring (*Bousfield, 1954*; *Crisp, 1968*; *Bouchard & Aiken, 2012*). On the studied coast, recruits of *S. balanoides* appear in May and June (*Ellrich, Scrosati & Molis, 2015*) along the full vertical intertidal range (*MacPherson & Scrosati, 2008*; *MacPherson, Scrosati & Chareka, 2008*). The recruited organisms reach reproductive maturity and adult size in the following fall (*Ellrich et al., 2016b*). In winter, the sea surface of the Gulf of St. Lawrence freezes extensively (*Galbraith et al., 2012*) and the ice causes physical disturbance in intertidal habitats as it moves with tides, winds, currents, and waves (*Scrosati & Heaven, 2006*; *Musetta-Lambert et al., 2015*). The ice melts before barnacle recruitment begins. As a result of ice scour, the macroscopic organisms occurring in high-intertidal habitats facing open waters (habitats where winter ice scour is intense) just before barnacle recruitment are almost exclusively adult barnacles scattered across otherwise bare rocky substrate. Benthic macroalgae and mobile consumers (*e.g.*, snails) remain virtually absent in such places during the barnacle recruitment season. Those organisms begin to appear at such elevations mostly after barnacle recruits have ceased to appear (*Scrosati & Heaven, 2007*). Thus, no interspecific interactions seemingly influence barnacle recruitment in those habitats, leaving adult barnacle density as an important potential predictor of barnacle recruitment.

We measured barnacle adult and recruit density at the lower part of the high intertidal zone, at an elevation of approximately 1.2 m above chart datum (lowest normal tide). During the second or third week of June between 2007 and 2016, we took digital pictures of 29–33 (depending on the year) 10 cm × 10 cm quadrats randomly positioned at that elevation spanning 20 m of coastline. With this approach, we were able to remove the influence of large-scale oceanographic factors on larval and recruit ecology (*Menge & Menge, 2013*) and, thus, to focus on small-scale influences at the patch (quadrat) level. We used the pictures of the quadrats to measure adult and recruit density on a computer. In June, barnacle recruits are easily distinguished from adults because recruits are only 1–2 mm in basal diameter (Fig. 1). Barnacle recruits were always abundant in the studied years (*Scrosati & Ellrich, 2016*), but adult barnacles were often absent in the quadrats between 2010 and 2016, preventing us from considering those years to meaningfully test our hypothesis. Therefore, we evaluated the hypothesis of this study using the datasets for 2007, 2008, and 2009.

For each quadrat, we calculated barnacle adult density by dividing the number of adult barnacles found therein by quadrat area (1 dm$^2$). To calculate recruit density, we divided the number of recruits found in a quadrat by the area that was available for larvae to settle (and, thus, for recruits to occur), which was quadrat area minus the area covered by adult barnacles because larvae did not settle on the shells of adult barnacles. For each quadrat, we measured the area covered by adult barnacles by analyzing the corresponding quadrat picture using ImageJ software. Under this analytical approach, if barnacle adults had no ecological effect on larval settlement and subsequent recruitment, there would be no relationship between adult density and recruit density (standardized as indicated above). A positive relationship would indicate attraction of larvae to adults, while a negative

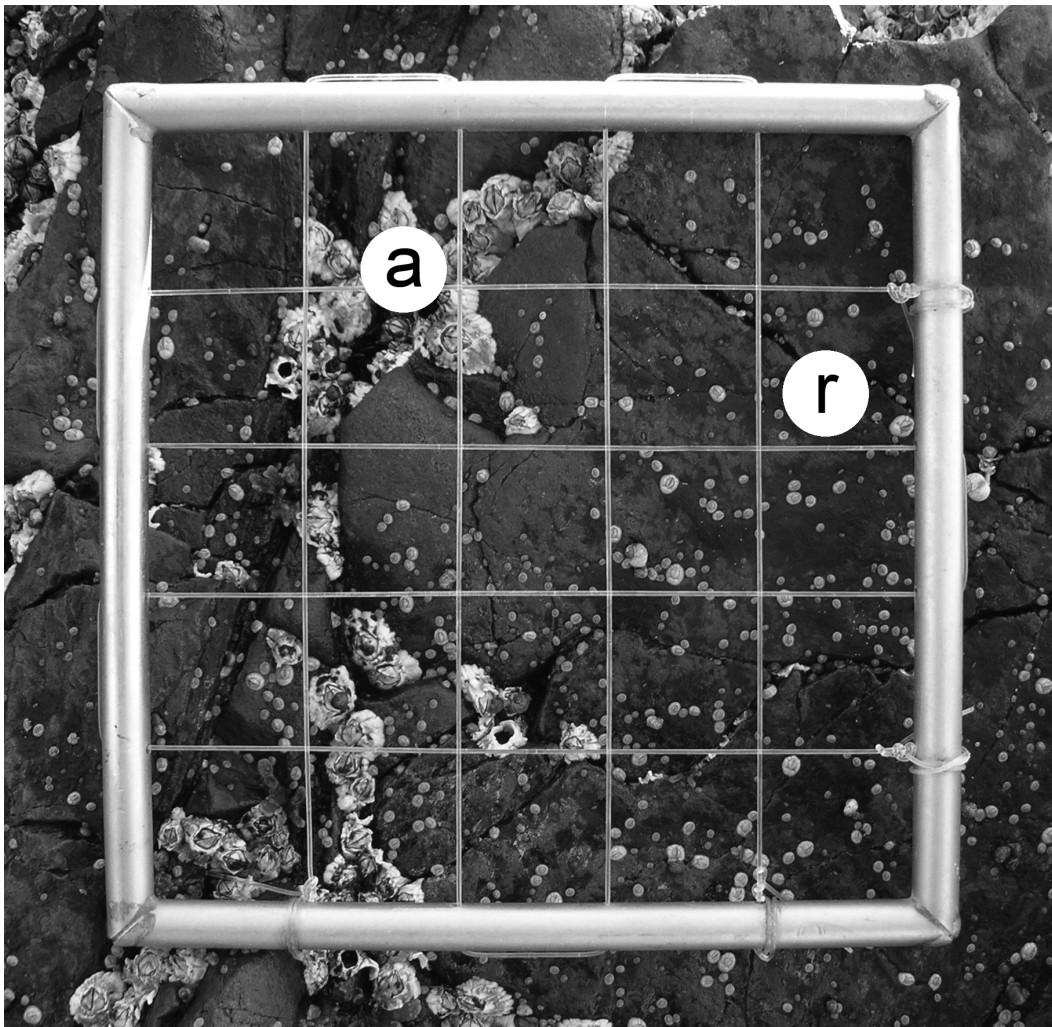

**Figure 1** **Typical view of a wave-exposed, high-intertidal habitat on the Gulf of St. Lawrence coast of Nova Scotia in June, showing barnacle adults (a) and recruits (r).** The sampling quadrat is 10 cm × 10 cm. Photograph by RA Scrosati.

relationship would indicate that larvae avoid places surrounded by increasingly high adult densities.

We tested our hypothesis by analyzing the 2007, 2008, and 2009 datasets separately. For each year, we fitted the data to a linear model, a power model, and a quadratic model, considering adult barnacle density as the independent variable and recruit density as the dependent variable. Then, we compared the three models using an information-theoretic approach. We considered linear and power models to evaluate potentially positive adult–recruit relationships. While a linear model would describe a constant rate of change of recruit density along the observed gradient of adult density, a power model would allow that rate to vary along that gradient, potentially making the model more realistic. We considered a quadratic model to evaluate the potential unimodal nature of the adult–recruit relationship. All models included an intercept to acknowledge the possibility that recruit
**Table 1** Summary information for the models describing the adult–recruit relationship in 2007, 2008, and 2009: a, b, and c are the parameters of the equations described in Methods, w is the weight of evidence for each model, and n is the number of surveyed quadrats.

| Model | a | b | c | Adjusted $R^2$ | AICc | w | Evidence ratio |
|---|---|---|---|---|---|---|---|
| **2007** ($n = 30$) | | | | | | | |
| Linear | 12.82 | 912.80 | – | 0.22 | 328.9 | 0.3077 | 1.8 |
| Power | 138.20 | 0.34 | 815.80 | 0.31 | 326.8 | 0.5607 | 1 |
| Quadratic | −0.34 | 25.44 | 877.90 | 0.25 | 329.7 | 0.1315 | 4.3 |
| **2008** ($n = 29$) | | | | | | | |
| Linear | 2.02 | 329.90 | – | 0.24 | 288.5 | 0.0536 | 14.2 |
| Power | 61.65 | 0.35 | 218.00 | 0.35 | 286.0 | 0.1872 | 4.1 |
| Quadratic | −0.04 | 7.65 | 228.80 | 0.40 | 283.2 | 0.7592 | 1 |
| **2009** ($n = 32$) | | | | | | | |
| Linear | 15.86 | 779.40 | – | 0.20 | 352.8 | 0.2887 | 1.9 |
| Power | 175.70 | 0.33 | 617.30 | 0.26 | 351.5 | 0.5529 | 1 |
| Quadratic | −0.49 | 31.49 | 736.10 | 0.20 | 354.0 | 0.1584 | 3.5 |

density could have non-zero values in the absence of adult barnacles. The models were: $N_R = aN_A + b$ (linear), $N_R = aN_A^b + c$ (power), and $N_R = aN_A^2 + bN_A + c$ (quadratic), where $N_R$ is recruit density and $N_A$ is adult density.

For each year, we compared the models based on their values of the corrected Akaike's information criterion (AICc). With such values, we calculated the weight of evidence for each model, which we used to identify the best model (the one with the highest weight of evidence) for each year. Then, for each year, we evaluated the plausibility of the best model relative to the two least favoured ones by calculating the corresponding evidence ratios between the weight of evidence for the best model and the weight of evidence for the two least supported models (*Anderson, 2008*). We calculated the adjusted squared correlation coefficient ($R^2$) for all models to determine the amount of variation in recruit density that was statistically explained by adult density (*Sokal & Rohlf, 2012*). We conducted these analyses with PRISM 6.0c for MacOS.

## RESULTS

In 2007 and 2009, adult barnacle density at the quadrat scale was never higher than 46 individuals dm$^{-2}$, while barnacle recruits occurred abundantly in all quadrats (Fig. 2). For both years, the best model describing the adult–recruit relationship was the power model (Table 1). For 2007, the power model was 1.8 times more plausible than the linear model and 4.3 times more plausible than the quadratic model while, for 2009, it was 1.9 times more plausible than the linear model and 3.5 times more plausible than the quadratic model (Table 1). The power model also explained a higher percentage of the observed variation in recruit density (31% in 2007 and 26% in 2009) than the other two models (Table 1).

In 2008, adult barnacle density exhibited a wider range than in 2007 and 2009. The highest value found in 2008 (129 individuals dm$^{-2}$) was almost three times higher than the

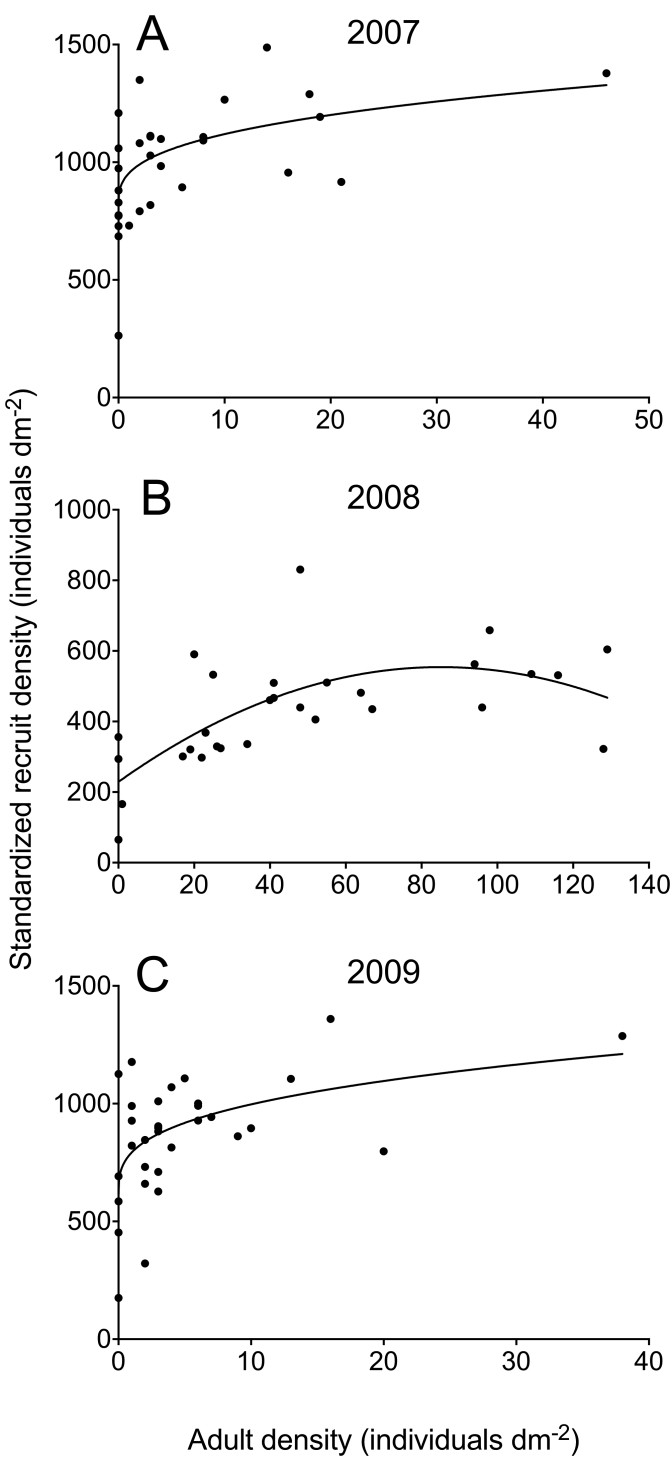

**Figure 2** **Relationships between standardized barnacle recruit density (scaled to the available area for recruitment in each quadrat) and the density of pre-existing adult barnacles (always scaled to quadrat area) in June 2007 (A), 2008 (B), and 2009 (C) for wave-exposed, high-intertidal habitats on the Gulf of St. Lawrence coast of Nova Scotia.** Each graph shows the model that best describes the adult–recruit relationship (see Table 1 for model parameters).

highest value found for the other two years. In 2008, barnacle recruits were also abundant on the shore (Fig. 2). The best model describing the adult–recruit relationship in 2008 was the quadratic one, which was 4.1 times more plausible than the power model and 14.2 times more plausible than the linear one (Table 1). For 2008, the quadratic model explained a higher percentage of the observed variation in recruit density (40%) than the other two models (Table 1).

## DISCUSSION

By monitoring a natural barnacle population spanning a wide range of adult density (in 2008), the predicted unimodal adult–recruit relationship was supported. When the available range of adult density covered only low to intermediate values (in 2007 and 2009), a positive relationship resembling the lower half of the unimodal relationship was found. This is a valuable outcome for two reasons. The first one is that this study provides an example of how the shape of a natural trend can be predicted from experiments that evaluated intraspecific interactions under different densities of adult barnacles separately, either under moderate densities (*Raimondi, 1988*; *Kent, Hawkins & Doncaster, 2003*; *Beermann et al., 2013*; *Ellrich et al., 2016a*) or high densities (*Bertness, Gaines & Yeh, 1998*; *Hooper & Eichhorn, 2016*). The second reason is that local-scale recruitment, a key driver of population persistence for sessile invertebrates (*Minchinton & Scheibling, 1993*; *Lathlean, Ayre & Minchinton, 2013*), is hereby shown to be linked to the abundance of adult conspecifics. At small spatial scales, adult barnacles generally do not contribute to local population persistence through the larvae they spawn, because pelagic larvae are taken elsewhere by water motion (*Caley et al., 1996*). However, the fact that settlement-seeking larvae are chemically and visually attracted to adults (regardless of where larvae come from) determines a positive adult–recruit relationship for moderate adult densities, which might be important for local population persistence. Habitat areas with higher adult densities, where population persistence may be less compromised and abundant adults may hinder recruit growth and survival, seem to stimulate larvae to settle elsewhere.

Using observed patterns of recruit density to infer patterns of larval settlement requires the assumption that post-settlement mortality until recruit counts are done is unimportant. This was seemingly the case for our system, because empty recruit shells (indicative of recruit mortality) were very rare at the time of our surveys (see, for example, Fig. 1). Mortality becomes important only later in the year, especially as the growing organisms experience thermal and desiccation stress during the summer low tides.

It is worth noting that an observational study done on the coast of New Brunswick, also in Atlantic Canada, reported a unimodal relationship for *Semibalanus balanoides* between recruit density and the percent cover of pre-existing adults (*Chabot & Bourget, 1988*). However, that study scaled recruit density to quadrat area. Given that larvae were noted (as in our study) to avoid settling on adults, that study suggested that recruit density decreased from intermediate to high values of adult cover because of the decreasing area available for larval settlement (*Chabot & Bourget, 1988*). As we scaled recruit density to the available area for settlement (which excludes adult barnacles), the present study in fact reveals that

avoidance of adults by settling larvae operates from intermediate to high values of adult abundance.

Immigration also represents the arrival of new organisms to a place, although it differs from recruitment because immigrants are normally actively moving juveniles and adults, not recruits. Thus, it is worth noting that a recent study using damselfish has experimentally demonstrated a unimodal relationship between local fish density and the density of conspecific immigrants (*Turgeon & Kramer, 2016*). The reasons for such a pattern seem also related to conspecific attraction at moderate densities and the potential for competitive interactions at high densities (*Turgeon & Kramer, 2016*).

The unimodal adult–recruit relationship for barnacles would likely hold mainly on shores where the supply of pelagic larvae to intertidal habitats is moderate. On shores subjected to a very high larval supply (e.g., because of mild water temperatures or high planktonic food supply), larvae quickly colonize preferred substrate and new larvae arriving to the shore eventually colonize less preferred areas also abundantly (*Bertness et al., 1992*). This suggests that, under very high larval supply rates, barnacle recruit density could be unrelated to the density of pre-existing adults. Seemingly in support of this notion, a study on Scottish shores subjected to an unusually high supply of *S. balanoides* larvae reported no linear relationship between the density of recruits and that of pre-existing adults (*Hansson et al., 2003*). However, the possible occurrence of nonlinear relationships was not evaluated in that study. In addition, the data examined in that study included density values from several intertidal elevations, which likely increased data variability because larval settlement rates increase towards lower elevations because of the longer immersion times (*Bertness et al., 1992*). It would thus appear useful to apply the methodology of the present study to evaluate on a suitable shore the notion that barnacle recruit density is unrelated to the density of pre-existing adults under very high rates of larval supply.

Finally, despite the variety of factors that influence barnacle recruitment at small scales (*Johnson & Strathmann, 1989*; *Bertness et al., 1992*; *Beermann et al., 2013*; *Coombes et al., 2015*; *Ellrich, Scrosati & Molis, 2015*; *Ellrich & Scrosati, 2016*), it is remarkable that a single factor (adult barnacle density) was able to explain 26–40% of the variation in recruit density observed on our shore. This may have been the case because the surveyed habitats exhibit a similar degree of wave exposure and substrate rugosity, composition, and slope. This was, in fact, the main reason to select this coast to address the objective of this study. In any case, a multifactorial field experiment could evaluate the relative explanatory ability of several small-scale factors acting simultaneously. This would be a profitable exercise, as the majority of studies have generally evaluated the influence of only one or two factors at a time (*Johnson & Strathmann, 1989*; *Beermann et al., 2013*; *Coombes et al., 2015*; *Ellrich, Scrosati & Molis, 2015*; *Ellrich & Scrosati, 2016*).

## ACKNOWLEDGEMENTS

We thank Donald Kramer, Augusto Flores, and an anonymous reviewer for constructive comments on an earlier version of this paper.

### Funding

This work was funded by a Discovery Grant (No. 311624) awarded to R. Scrosati by the Natural Sciences and Engineering Research Council of Canada (NSERC) and by a Ph.D. scholarship (No. D/10/47054) awarded to J. Ellrich by the German Academic Exchange Service (DAAD). The funders had no role in study design, data collection and analysis, decision to publish, or preparation of the manuscript.

### Grant Disclosures

The following grant information was disclosed by the authors:
Natural Sciences and Engineering Research Council of Canada (NSERC),
Discovery Grant: 311624.
German Academic Exchange Service (DAAD), Ph.D. scholarship: D/10/47054.

### Competing Interests

The authors declare there are no competing interests.

### Author Contributions

- Ricardo A. Scrosati conceived and designed the experiments, performed the experiments, analyzed the data, contributed reagents/materials/analysis tools, wrote the paper, prepared figures and/or tables, reviewed drafts of the paper.
- Julius A. Ellrich performed the experiments, analyzed the data, reviewed drafts of the paper.

### Data Availability

The raw data was supplied as a Data S1.

### Supplemental Information

Supplemental information for this article can be found online at http://dx.doi.org/10.7717/peerj.3444#supplemental-information.

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
