# Peer review of "Unimodal relationship between small-scale barnacle recruitment and the density of pre-existing barnacle adults"

_PeerJ, doi:10.7717/peerj.3444_

## Round 0.1 · original submission · Minor Revisions

I have received two reviews of your manuscript, both of which indicate that it would be suitable for publication following minor revisions, and I concur with this view. Overall, the manuscript is interesting, succinct and well written.

The reviewers do have some suggestions for additional clarity and completeness. Please consider all comments carefully and make appropriate changes or explain why you believe it is not appropriate to do so. I have some additional suggestions, which you may treat as a third review.

Editor's Comments
Conceptual issues:
Potential for spurious correlation: In the coral reef fish literature, there has recently developed a recognition that observed negative density dependence could be artefactual. If larvae settle in an area independent of density but settler density is divided by adult density, a negative relationship must result. This has been called pseudo-density dependence. The relevance to your study is that if barnacles settled in a 10 x 10 cm area independent of adult density but recruit density is scaled to available space there will be a positive trend with adult density, even if settlers do not respond to adults. This will exaggerate any positive trend and diminish any negative trend in your data. Looking at the range of available space values in your data, it appears unlikely that this is a serious artifact. Nevertheless, it is worth considering. I am sensitive to this issue because I recently published, with my former doctoral student Katrine Turgeon, a paper on density dependence of immigration in a territorial coral reef fish (which also showed unimodal/hump-shaped relationships), and we had to address this issue. Our paper (Turgeon & Kramer 2016 PLoS ONE 11(6): e0156417) provides references [last few lines of p.4]. Your conclusions might be strengthened by repeating your analysis using recruits per quadrat as the dependent variable to confirm that the patterns hold.

L21 (and elsewhere L153-156). Your wording seems to imply that the active agent is adult barnacles that are selected to attract juveniles to maintain population persistence. Although you may not intend this, it raises issues of group selection, spatial limits of the barnacle populations which are probably much greater than the scale of habitat selection in your study, and whether selection would act on adults to attract juveniles or on juveniles to select habitat with more favorable adult densities (or both). While there is literature showing that juveniles can alter their response to different adult densities, I wonder whether any studies show that adults adjust their attractiveness to their own population sizes. You might reconsider whether the wording correctly reflects your understanding of the system.

L25. 'Stimulated to settle elsewhere' also seems somewhat unclear. Would 'stimulated to avoid high conspecific densities' be clearer?

Wording, grammar and style:
L7. Is 'step' the right word here? Process? Stage?
L14 (and elsewhere in the manuscript: L60, 309, 311). I did not know what you meant by 'referred to' when I read the abstract. Is this a standard term? Would 'scaled to' or 'per unit of' be clearer to readers?
L17. I think it would be clearer to say '. . . the relationship . . . was positive and resembled . . .'.
L18. No space between number and % sign.
L26. 'because' would be clearer than 'as' in this context to avoid implication of simultaneous events.
L72. Do you mean 'broods' rather than 'breeds'? In most animals, mating is considered part of breeding.
L89. 'During' not 'at'.
L102. Somewhere in your Methods, you should refer to the supplementary raw data and its location. I would also suggest that you make the years more evident. They do not appear in column heads until selected. When I first looked at the supplementary data, I did not recognize that you had data for all three years in successive columns. The years could come first in the column heads, or you add a column for year and present all the data in successive rows of the same three columns.
L164. Is 'notion' the best term for this suggestion?

·

Basic reporting

All aspects are met.

Experimental design

The experimental design seems adequate, but the authors need to clearly state what is the spatial scale of their study – this is crucial because the relative importance of conspecific interactions (which I understand are the focal mechanisms of this study) may be high very locally but low if plots span a coastline of hundreds of meters or more. In this case, the authors should test for spatial autocorrelation of both adult and recruit density (if possible, of course) – their interpretations would gain more credibility if there is none.

I specify below where in the manuscript the authors should address the above issues:

Line 91: What was approximately the average distance between plots? What was the length of the sampled coastline? This information is critical since processes responsible for spatial recruitment patterns may dramatically change with scale (say from tens of m to unit km).

Validity of the findings

This is an interesting study that addresses a central question on the dynamics of space-monopolizing invertebrates. When adult populations become very dense, there may be spatial limitation for new recruits and larval responses at settlement may well be adaptive – in this context, the present study supports positive and negative chemical mediation of larval settlement (or/and early post-settlement survival) at low and high adult densities, respectively, which may ultimately homogenize barnacle density and reduce mortality of settling larvae and early juveniles.

I found, however, that authors had not convinced readers that processes other than intraspecific interactions (such as larval supply and predation, which could potentially be correlated to adult density) are not relevant in their study system. I point out below ways in which the authors may countervail this weakness.

Unlike the first three sections of the manuscript, the discussion is a bit confusing and some more work is needed, mostly in the interpretation of results which I found not very parsimonious.

I specify below where in the manuscript the authors should address the above issues:

Introduction

Lines 49-52: Chemical mediation of larval settlement should be a key issue in this section. I suggest expanding a bit these lines. Is there any evidence of adult density thresholds affecting the behavior of competent larvae? Any effects on very early post-settlement mortality, that may not be detected at usual sampling frequencies (weekly, fortnightly)?

Lines 53-54: This might be plausible, provided that (i) populations are not limited by the supply of settlers, and (ii) nearshore cyprid abundance is not correlated to local adult standing stock. How often are these conditions met? The authors may wish to restrict this to some general environmental / oceanographic conditions (?) – Please elaborate here a bit more.

Lines 145-151: This is confusing. (i) I do not agree with authors if by ‘predicted’ (line 146) they mean that recruit density can be anticipated by estimating the density of adults at any given time. It is true that ‘predicted’ recruit densities in 2007 and 2009 are reasonably similar, but they are very different form the values found in 2008. For instance, at an adult density of 20 ind.dim-2, the model would return recruit densities around 1,100 – 1,200 ind.dm-2 in 2007/2009 but only 300-400 ind.dm-2 in 2008. (ii) Recruit density was actually correlated to adult density for the whole range. Any eventual intraspecific interactions (line 148) would therefore underlie the full response, not only at ‘high adult densities’. Provided the authors have convinced readers that local adult reproduction cannot be coupled to local recruitment, I would encourage the authors to elaborate more on possible divergent larval responses (low / moderate vs. high adult density) to adult conspecific cues (aligned to what was already written in lines 154-157).

Lines 151-153: Be careful here – several studies have found that self-recruitment may be very important for different marine invertebrates and (mostly) reef fish. The review by Caley and colleagues is an important one, but most of the evidence of self-recruitment was published after 1996 and therefore not addressed at that time. I would accept this statement if by ‘local’ authors state, say, tens of meters, but would be skeptical if there is substantial variation of adult density at a scale of hundreds of meters or more (if authors have measured distances in sequential plots a test for spatial autocorrelation could be provided).

Lines 170-172: The results shown in this study support the notion that relationships still hold, as long as recruit densities over 1,000 ind.m-2, as reported in 2007 and 2009, do suggest very high larval supply.

Additional comments

Other comments not addressed in sections 1-3.

General comments:

Lines 83-85: This is confusing and perhaps needs redrafting. If I got it right, the authors claim that macroalgae and invertebrate consumers are absent because they recruit or move to the upper intertidal only in late spring / summer - However, this is exactly when barnacle recruitment takes place (?), as stated above in the same paragraph (lines 73-75). This is important, as long as predation maybe an important process (as pointed out in lines 41-42), and the access of consumers to plots was not controlled.

Line 104: The quadratic model does not translate well to the ‘unimodal’ pattern, as referred in other sections of the manuscript, including the title. I would suggest changing the ‘unimodal’ term accordingly (unless the authors decide to change the approach and test the goodness of fit of theoretical frequency distributions to data, which I believe they will not).

Minor issues:

Title: in a barnacle population?

Line 60: In these latter conditions?

Lines 60-61: with larvae swimming away from crowded habitats? A very brief explanation need to be inserted here.

Table 1: The evidence ratio is not a very intuitive measure (i.e. higher ratios mean lower evidence for respective models). Perhaps this metric could be briefly explained in this table heading.

Reviewer 2 ·

Basic reporting

The manuscript is well presented and uses clear unambiguous language throughout. I think it could be improved slightly by incorporating the following reference, which is particularly relevant to the study:

- Minchinton TE & Scheibling (1993) Free space availability and larval substratum selection as determinants of barnacle population structure in a developing rocky intertidal community. MEPS 95: 233-244.

Experimental design

Methods are described with sufficient detail such that it could be replicated elsewhere. However, whilst the research question is well defined, I think the authors could improve the manuscript by better identifying how they are filling a particular knowledge gap. My initial thoughts were that this question has previously been addressed. I may be wrong, or in fact, not totally across the topic. A single sentence or two in the intro is all that is needed to highlight how this improves our understanding.

Validity of the findings

I believe that the data presented by Scrosati & Ellrich is robust and statistically sound. I particularly liked their use of AIC comparisons to assess which model was most appropriate.

Discussion was also well presented with authors offer some good explanations as to why recruitment in 2007 and 2009 was linear whilst it was unimodal in 2008.

Additional comments

This work presents a short report that addresses an important question in benthic marine ecology i.e. how do adult densities influence recruitment success/population replenishment? The authors take advantage of their study system to address this question through a series of field observations where they correlate the number of newly established barnacle recruits with the density of the surrounding adult barnacles. Whilst this is not the first time this question has been addressed within the literature (see Lathlean et al. 2012, 2013) I believe the results provide valuable new insights into the nature of this relationship. Saying that, I would like to know how authors tried to standardise the various other small-scale factors that are known to influence barnacle settlement and recruitment, namely rugosity/topography. I appreciate that the nature of field experiments generally do not permit all these various factors to be accounted for. But based on the photograph in Figure 1 the small-scale topographic complexity seems to be quite high and it may be interacting with patterns of adult density/barnacle recruitment. The authors do suggest in the discussion (L177-178) that the high degree of correlation (26-40%) between just a single factor (i.e. adult density) and barnacle recruitment may be due to habitats displaying similar "wave exposure and substrate rugosity, composition, and slope." However, it might be useful to say something of this nature in the Methods section.

---

## Round 0.2 · accepted · Accept

The manuscript is now suitable for publication, with a small number of changes, previously agreed to in email communication, indicated by inserted comments to replace highlighted text. It is a nice study, and I am happy to have been involved with its review.